# Recurrent Emergency Department Users: Two Categories with Different Risk Profiles

**DOI:** 10.3390/jcm8030333

**Published:** 2019-03-09

**Authors:** Ksenija Slankamenac, Meret Zehnder, Tim O. Langner, Kathrin Krähenmann, Dagmar I. Keller

**Affiliations:** Emergency Department, University Hospital Zurich, Raemistrasse 100, CH-8091 Zurich, Switzerland; meret.zehnder@usz.ch (M.Z.); tim.langner@usz.ch (T.O.L.); kathrin.kraehenmann@usz.ch (K.K.); dagmar.keller@usz.ch (D.I.K.)

**Keywords:** repeated and frequent emergency department visitors, recurrent emergency department visits

## Abstract

Recurrent emergency department (ED) visits are responsible for an increasing proportion of overcrowding. Therefore, our aim was to investigate the characteristics and prevalence of recurrent ED visitors as well as to determine risk factors associated with multiple ED visits. ED patients visiting the ED of a tertiary care hospital at least four times consecutively in 2015 were enrolled. Of 33,335 primary ED visits, 1921 ED visits (5.8%) were performed by 372 ED patients who presented in the ED at least four times within the one-year period. Two different categories of recurrent ED patients were identified: repeated ED users presenting always with the same symptoms and frequent ED visitors who were suffering from different symptoms on each ED visit. Repeated ED users had more ED visits (*p* < 0.001) and needed more hospital admissions (*p* < 0.010) compared to frequent ED users. Repeated ED users visited the ED more likely due to symptoms from chronic obstructive pulmonary diseases (*p* < 0.001) and mental disorders (*p* < 0.001). In contrast, frequent ED patients showed to be at risk for multiple ED visits when being disabled (*p* = 0.001), had an increased Charlson co-morbidity index (*p* = 0.004) or suffering from rheumatic diseases (*p* < 0.001). A small number of recurrent ED visitors determines a relevant number of ED visits with a relevance for and impact on patient centred care and emergency services. There are two categories of recurrent ED users with different risk factors for multiple ED visits: repeated and frequent. Therefore, multi-professional follow-up care models for recurrent ED patients are needed to improve patients’ needs, quality of life as well as emergency services.

## 1. Introduction

Swiss healthcare costs continued to rise to a total of 80.7 billion CHF in 2016, which was 3.8% more compared to 2015 [1]. It significantly burdens the gross domestic product and needs to be reduced. Healthcare costs are also burdened by the continuous rise of visits to emergency departments (ED). In addition, this rise of visits results in ED overcrowding. The subgroup of patients who repeatedly use the ED is still a small proportion of ED patients but extensively use ED resources, causing ED overcrowding and decreasing the quality and efficiency of emergency care [2,3,4,5]. These recurrent ED visits also charge costs, insurance companies and the healthcare system significantly more compared with patients adequately managed by primary care providers [6,7,8]. While EDs were primary designed for “real” emergency care, recurrent visitors to the ED due to various medical problems constitute a considerable proportion of the total number of ED visits [9,10]. 

Many different definitions for “ED recurrence” have been reported [11,12,13] but most articles consistently defined at least four ED visits during a one-year period as a recurrent ED visitor [14,15,16,17,18,19].

Mainly, many health problems of recurrent ED visitors are triaged by medical personnel as less or non-urgent, for example, according to the emergency severity index (ESI) levels 4 or 5 [20] and could therefore be well managed in primary care settings [21,22]. A study in the U.S. showed in 2005 that, surprisingly, 93% of recurrent visitors had a primary care provider for medical issues [23]. But lack of satisfaction with the choice and performance of primary care physicians as well as unmet medical needs led to repeated ED visits [15,24,25]. Prevalent medical reasons for repeated visits in US EDs were sprains, back problems and headaches [6]. An older publication from 1985 showed that ED users with multiple visits had significantly more psycho-social problems such as feelings of loneliness, living alone, receiving disability pension, having contacts with social insurance agencies, a high number of sick absenteeism from work and alcoholism [26]. Even though there are certainly patients using the ED multiple times for low-acuity symptoms, those ED visitors were often sick patients with chronic diseases associated with increased hospital admission rates due to exacerbation of the underlying diseases and/or co-morbidities [3,6,23]. 

In Switzerland, recurrent ED visits have not been entirely investigated. Therefore, we investigated the number and proportion of recurrent ED users and evaluated characteristics, causes and risk factors within the recurrent ED visitors.

## 2. Experimental Section

### 2.1. Study Design and Setting

We enrolled all patients of the ED of a tertiary care hospital from 1st January 2015 to 31st December 2015 who had visited the ED at least four times during a one-year period. Those patients were called recurrent ED visitors. Even though there has not been a standardized definition for “recurrent”, the majority of studies defined at least four visits per year as a valid threshold [14,15,16,17,18,19]. 

The following patients were excluded: patients younger than 18 years old, patients who visited the ED less than four times during the defined one-year period and/or patients who visited the ED for planned follow-up examinations or treatments. 

The study was approved by the local ethics committee (BASEC-Nr.2016-00343). 

### 2.2. Endpoints

The primary endpoint was the number and subsequent proportion of recurrent visits and visitors in the ED within a one-year period. 

The following parameters were investigated as secondary endpoints within the recurrent ED population: characteristics, number of hospital stays, number of presentations during the day- (08 am to 5 pm), evening- (05 pm to 11 pm) and night-shifts (11 pm to 08 am) as representative of the “around-the-clock availability” and “one-stop one-shop” mentality as well as risks for repeated or frequent ED visits. Secondary endpoints evaluated whether multiple ED visits were caused by the effect of “around-the-clock availability” of EDs, “one-stop one-shop mentality” of patients, convenience, psychosocial aspects or by sicker patients with chronic diseases. “One-stop one-shop” mentality has been increasingly seen in ED patients. They wished to solve all their health problems and needs in “one stop” by “shopping” multiple emergency services for multiple health problems during one ED visit. 

### 2.3. Assessment of Other Study Variables

Additional to the endpoints, we retrospectively ascertained clinical and demographic data from the in-hospital digital clinical information system in order to characterize the enrolled population. We assessed triage at admission by the emergency severity index (ESI) [27]. The severity and urgency of ED visits were recorded using the ESI. ESI 1 is describing a life-threatening emergency, ESI 2 a high-risk situation and ESI 3 a medical situation with the need for several medical resources. In ESI 4 and 5, no medical resource was needed [20,28,29]. Furthermore, we assessed age, gender, primary diagnosis, co-morbidities (e.g., coronary heart disease, chronic kidney failure, underlying malignant disease etc.), Charlson co-morbidity index (CCI) [30]—which is a representative for the severity of multiple co-morbidities and predicts 10-year survival—regular medication, drug consumption, suicide attempts in the past, domestic violence in the history, ED self-admittance or by paramedics or external doctors, presence of a general physician (GP) and social data such as homeless (yes/no), housing (living alone, living with others, supervised living or unknown), marital status (single/married), divorced (yes/no) and widow/-er (yes/no), children (yes/no) as well as professional status (differentiating between being employed, unemployed, in training, retired or disabled). 

### 2.4. Statistical Analysis

As the first step, we expressed the distribution of variables using means and standard deviation for normally distributed data and medians and interquartile ranges for non-normally distributed data. We tested the dataset for normality with the Kolmogorov-Smirnov test. Categorical data were presented as frequency. In the case of missing data, we used the multiple imputation method, a statistical technique for analysing incomplete data sets.

The primary endpoint (number of recurrent ED visits) as well as the characteristics and causes for recurrent ED visits were presented as proportions.

To identify risk factors for repeated and/or frequent ED visits, a stepwise backward regression analysis was performed. We also performed a subgroup analysis on senior recurrent ED patients, 70 years or older, by using stepwise backward regression models to identify risk factors between repeated and frequent senior ED visitors. 

We compared the primary as well as secondary endpoints between the repeated and frequent groups using simple and multivariable logistic or linear regression models adjusting for potential confounders. Furthermore, we focused our regression analysis within the repeated or frequent populations and discriminated within these groups between multimorbid patients (CCI ≥ 4) and less sick patients (CCI < 4) [31] and those ED patients that had a GP or not.

For all results, we reported point estimates, 95% confidence intervals and *p*-values (<0.05 considered significant). We performed statistical analyses using the statistical program STATA SE (version 15, Stata Corp., College Station, TX, USA). 

## 3. Results

In 2015, 40,496 patients visited the ED. After excluding patients according to the exclusion criteria, 27,998 ED patients resulted who performed in total 33,335 primary ED visits whereof 1921 ED visits (5.8%) were performed by 372 recurrent visitors (1.3%). 

We identified for the first time two different groups within the recurrent visitors: ED visitors who presented in the ED due to “different” symptoms and health problems at each presentation, so-called “frequent” ED visitors. These 213 frequent ED patients performed 1016 ED visits during the one-year observation. The second group of recurrent ED patients always presented due to identical symptoms, so called repeated ED visitors. These 159 repeated ED patients were responsible for 905 of those 1921 ED visits. In three patients, we had to discuss the allocation. All three patients had more than six ED visits due to identical health problems but each of them showed one ED visit due to different symptoms to the previous visits. Since the majority of the ED visits were in all three cases due to identical symptoms, we therefore decided to group them as repeated ED users.

The non-recurrent ED population (*n* = 27,626) was an average of 45 years (SD 19) old, 54% of them were male. After tests and first treatment in the ED, every fourth patient (25.7%) needed to be admitted to the hospital. 43.4% of patients visited the ED due to consequences of an injury or trauma whereas the remaining 56.6% had medical problems including poisoning, drugs or alcohol intoxications and psychological disorders. Most of the non-recurrent patients (63.4%) were triaged into the ESI category 3, followed by 29.6% as ESI 4, 6.5% as ESI 2 and 0.5% as ESI 5.

All patients’ characteristics and co-morbidities are presented in Table 1. The recurrent ED patients in general were an average of 50 years of age (SD 18.5 years) and the majority were male (58.6%). Almost 40% of the patients were multimorbid showing a Charlson co-morbidity index (CCI) of four or higher. The four leading co-morbidities in this ED population were arterial hypertension (39.5%), mental disorder (32.5%), chronic kidney failure (23.1%) and rheumatic disease (21.8%). Fifty-eight patients (15.6%) showed regular drug abuse in the patient history and 43 patients (11.6%) had reported a suicide attempt in the past. 

Focusing on the main group differences between repeated and frequent ED visitors (Table 1), repeated visitors had fewer co-morbidities (CCI ≥ 4: 29.6% vs. 46.9%) but suffered more often from a mental disorder (46.5% vs. 22.1%) and showed a higher incidence of suicide attempts in the past (15.1% vs. 8.9%) than frequent ED users did. In contrast, frequent ED visitors were sicker and suffered more often from chronic diseases such as arterial hypertension (43.2% vs. 34.6%), chronic kidney failure (28.6 vs. 15.8%), rheumatic (31% vs. 9.4%) or malignant diseases (23% vs. 12.6%) compared to repeated visitors (Table 1). 

Social parameters are shown in detail in Table 2. Most ED patients had a general physician (GP) (77.7%), were married (45.7%) and half of them had children. Almost every seventh recurrent ED patient was unemployed (15%). Similar results were shown when differentiating between repeated and frequent users except that repeated ED visitors were more often singles (41.5% vs. 31%).

Patients were triaged at ED admission by the ESI [20]. 1346 of 1921 recurrent ED visits (70.1%) were triaged as ESI level 3 (Table 3). Almost half of the patients visited the ED during office hours. Three-quarters of recurrent ED patients presented between 08 am and 11 pm in the ED. The majority presented in the ED by self-admittance (Table 3). 

### 3.1. Leading Symptoms of Repeated and Frequent ED Visitors

The two leading symptoms for repeated ED presentations were mental (28.3%) and gastrointestinal (21.4%), followed by cardiac (12.6%), neurological (10.7%) and respiratory (8.2%) symptoms (Table 4). 

Symptoms of frequent ED patients were caused by gastrointestinal (17.3%), trauma (13.4%), mental (10.3%), neurological (9.5%) or cardiac (9.4%) disorders (Table 5). 

### 3.2. Repeated ED Visitors

Repeated ED visitors had significantly more ED visits (3 (IQR 2–5) versus 3 (IQR 2–4); adjusted difference 0.9, 95% confidence interval (CI) 0.4–1.3, *p* < 0.001) and needed more inpatient treatments (2 (IQR 0–4) versus 2 (IQR 0–3); adjusted difference 0.5, 95% CI 0.1–0.8, *p* < 0.010) compared to frequent ED patients. 

They were more likely to present in the ED due to symptoms from acute respiratory (OR 6.6, 95% CI 2.9–15.0, *p* < 0.001) and mental disorders (OR 3.9; 95% CI 2.2–6.8, *p* < 0.001). A subsequent subgroup analysis in patients with acute respiratory disorders was due to a small sample of 13 patients. Table 6 shows differences between the repeated ED patients suffering from symptoms of an acute mental disorder when presenting in the ED compared to the remaining repeated ED population. Repeated ED users had significantly more ED visits (*p* = 0.003), more hospital admissions (*p* = 0.011) and presented more often after 5 pm (*p* = 0.008) due to symptoms of acute mental disorders (Table 6). Additionally, those patients who presented due to symptoms of acute mental disorders had an increased risk for concomitant intoxication when presenting in the ED (*p* = 0.010), to have undergone a suicide attempt in the past (*p* < 0.001), being single (*p* = 0.010) and unemployed (*p* < 0.001) (Table 6).

A further subgroup analysis by a multivariable linear and logistic regression analysis showed that multimorbid repeated ED patients (CCI ≥ 4) did not have more ED visits (*p* = 0.14) or number of hospital admissions (*p* = 0.12) compared to less sick ED patients (CCI < 4). Sicker repeated ED patients suffered significantly less likely from acute mental disorders when presenting in the ED (adjusted OR 0.3, 95% CI 0.1–0.9, *p* = 0.039). 

Although 79.2% of the repeated ED patients had a GP there were not fewer ED visits in total or during the office hours when the GP practice was open.

### 3.3. Frequent ED Visitors

The frequent ED populations showed the following risk factors for multiple ED visits: being disabled (OR 3.0, 95% CI 1.6–5.7, *p* = 0.001), increased Charlson co-morbidity index ≥4 (OR 2.4, 95% CI 1.3–4.5, *p* = 0.004) and suffering from an underlying rheumatic disease (OR 2.6, 95% CI 2.6–10.8, *p* < 0.001). Being homeless (OR 3.6, 95% CI 0.9–14.6, *p* = 0.076) or unemployed (OR 2.0, 95% CI 0.95–4.0, *p* = 0.068) showed a trend for an increased risk of multiple ED visits in frequent ED users. Frequent ED patients with an increased CCI ≥ 4 needed more hospital admissions (2 (IQR 2–4) versus 1 (IQR 0–2); adjusted difference 1.0, 95% CI 0.5–1.6, *p* < 0.001) but presented less often during night shifts (adjusted difference −0.2, 95% CI −0.6 to 0.3, *p* = 0.040) compared to less sick patients (CCI < 4). 

Although more than three-quarters of the frequent ED patients had a GP (76.5%) there were not fewer ED visits in total or during the office hours when the GP practice was open. 

### 3.4. Subgroup of Recurrent Senior ED Patients

Since senior patients showed more co-morbidities and an interaction between increased age (≥70 years.) and CCI was found, we performed a subgroup analysis in the recurrent senior ED population. We identified 65 recurrent ED seniors who made in total 316 ED visits during the defined one-year period. Of these, 40 frequent ED seniors presented 195 times whereas 25 repeated ED seniors were responsible for 121 ED visits. The five leading reasons for seniors presenting recurrently in the ED were cardiac disorders (14.9%), postoperative complications (13.3%), respiratory (12%), gastrointestinal (11.1%) and neurological disorders (10.1%). In a stepwise backward regression analysis, recurrent ED visits by seniors were strongly associated with weekend presentations (*p* < 0.001). 

Repeated ED visits were associated with female gender (OR 5.9, 95% CI 1.4–24.8, *p* = 0.016) and underlying coronary heart disease (OR 4.6, 95% CI 1.02–21.1, *p* = 0.046), underlying malignant disease showed a trend to an increased risk (OR 4.5, 95% CI 0.8–24.1, *p* = 0.081). 

Predictors of frequent ED visits in seniors were increased age (>75 years.) (OR 1.2, 95% CI 1.1–1.5, *p* = 0.004) and chronic kidney failure (OR 4.6, 95% CI 1.3–16.8, *p* = 0.021).

## 4. Discussion

Of 33,335 primary ED visits in a one-year period, almost 6% (*n* = 1921) were performed by recurrent ED visits. We distinguished two different categories of recurrent ED patients by presenting different risk factors for repeated and frequent ED visits. Additionally, repeated ED patients generated significantly more ED visits than frequent users. The main reason for multiple ED visits in repeated visitors were symptoms from mental disorders whereas in frequent ED users mainly gastrointestinal symptoms caused the high number of ED visits. Thus, repeated ED patients with symptoms of mental disorders had significantly more ED visits, hospital admissions and presented more often after office hours. Furthermore, they had an increased risk for repeated ED visits due to social factors such as being single or unemployed. In contrast, frequent ED patients were shown to be at risk for multiple ED visits when being disabled, multimorbid or suffering from an underlying chronic disease. 

The prevalence of recurrent ED visitors varies widely due to different definitions of “frequent ED use” in the literature. Definitions of “frequent use” vary from two to 12 visits per year [6,32,33,34,35,36]. The threshold for the definition of a recurrent ED visitor was most commonly four or more visits per year [3]. Therefore, we defined at least four visits/year as a threshold for recurrent ED visits. The prevalence of recurrent ED users varies from 1 to 8% of all ED patients and 17% to 28% of all ED visits [3,15,34]. This current study showed a prevalence of 1.3% of recurrent ED patients and was in the range of the literature [3,34]. Counting ED visits, our study reported a lower frequency of recurrent ED visits (5.8%) [3,15,16,34]. There are two possible explanations. First, this lower frequency of ED visits might be caused by a very low number of heavy ED users using the ED more than 15 times per year compared to the literature [35,37,38]. Only sixteen patients showed ten or more ED visits per year and beneath these heavy users, only three patients had 15 or more visits/year. Second, the lower frequency of recurrent ED visits might be due to many surrounding EDs and hospitals. It is known that this group of ED patients prefer to visit multiple hospitals in the same period [6,14,34,39]. Thus, the reported prevalence of recurrent ED visits is probably underestimated and likely to be much higher in reality. Therefore, there is a need for a nationwide digital patients’ registry to receive more follow-up information and an overall view of all hospital and ED visits, medications and diagnosis.

According to our knowledge, none of the articles in the literature had differentiated within recurrent ED patients. We identified two different categories of recurrent ED users in the current study. Thus, we were also able to identify different risk factors for repeated and frequent ED visits that may have major consequences in further emergency services and decision-making. The previous literature has only described that recurrent ED visitors were a heterogeneous group of patients with an increased risk for ED visits due to substance abuse, poor mental health, mental disorders, greater social problems and poor physical health [6,11,15,38,40,41,42]. Most of the former articles were descriptive and rarely evaluated risk factors [34,37]. We therefore assume that previous risk analyses were hard to perform due to the heterogeneous population. Thus, we showed that symptoms of mental disorders, social factors such as being unemployed and being single were only significantly associated with the repeated ED group of patients, whereas poor physical health represented by chronic disease and multimorbidity was strongly associated with frequent ED users. This novel perception of two categories repeated and frequent ED patients with different risk profiles are important findings for future approaches to emergency services and patient-centred care within the recurrent ED users. To reduce (over)crowding in the ED, the follow-up care between the two groups of ED patients must be completely different. While repeated ED patients need support by way of case management, frequent ED patients may benefit more from integration in the primary care setting. 

It is known that recurrent ED visitors have more hospital admissions than non-recurrent ED users [3,6,11,40]. Focusing on our results, we showed for the first time within the recurrent patient population that repeated ED users had significantly more ED visits and needed more in hospital treatments than frequent ED visitors did. 

There are only a few articles in the literature reporting that recurrent ED patients suffered more often from poor health [11,15,25]. A systematic review showed in 2010 that sicker recurrent ED patients more often needed a hospital admission than less sick patients [3]. First, none of these articles used the Charlson co-morbidity index to objectively assess terms such as “poor health” or “sick.” We defined objectively multimorbid patients by the Charlson co-morbidity index (CCI) (≥4) [31] and showed that only multimorbid frequent ED patients had significantly more hospital admissions. In contrast, multimorbid repeated ED patients did not have more hospital admissions. Another novel finding in our analysis is that multimorbid frequent ED users presented less often during night shifts whereas recurrent ED users were more common during evening and night shifts [17,43]. One reason for that difference is that multimorbid frequent ED visitors may use the ED as a GP practice and therefore present less often during night shifts. Another reason may be that they suffer from less heavy and less urgent symptoms so that they do not need to visit the ED during the night. Furthermore, our study showed that neither repeated nor frequent multimorbid ED patients had more ED visits than less sick ED users. 

### Limitations and Strength of the Study

Our study has several limitations. First, it is a single-centre study in a city with a tight net of other EDs and hospitals around our tertiary care hospital. Therefore, it is likely that the number of recurrent ED patients is much higher than reported in this study because they are likely to be seen at multiple EDs. A second possible limitation might be the retrospective design but due to the digital clinical information system full of information, we had a very low number of missing data. In the case of missing data, we performed a multiple imputation, a statistical technique for analysing incomplete data sets. We turned this negative effect to our strength by using modern and appropriate statistical techniques. A further strength was the multivariable regression analysis to adjust for possible confounders and to reduce the bias in this retrospective study. 

## 5. Conclusions

Almost 6% of ED visits were made by a small number of recurrent ED visitors (1.3%) with important relevance for and impact on patient centred care and emergency services. There are two categories of recurrent ED users: repeated and frequent. This novel perception of two categories of ED patients with different risk profiles and characteristics are important findings for future approaches. Therefore, innovative and multi-professional follow-up care models as well as a nationwide patients’ registry for recurrent ED patients are needed in the future to improve patients’ needs, quality of life, patient-centred care and emergency services. 

## Figures and Tables

**Table 1 jcm-08-00333-t001:** Characteristics and co-morbidities of the recurrent emergency department (ED) visitors.

	All Recurrent ED Visitors *N* = 372	Frequent ED Visitors *N* = 213	Repeated ED Visitors *N* = 159
Age, years	50.3 (18.5)	51.2 (19.1)	49.2 (17.6)
Sex (female), *n* (%)	154 (41.4%)	85 (39.9%)	69 (43.4%)
Charlson co-morbidity index	2 (0–6)	3 (0–6)	1 (0–4)
<4, *n* (%)	225 (60.5%)	113 (53.1%)	112 (70.4%)
≥4, *n* (%)	147 (39.5%)	100 (46.9%)	47 (29.6%)
Arterial hypertension, *n* (%)	147 (39.5%)	92 (43.2%)	55 (34.6%)
Mental disorders, *n* (%)	121 (32.5%)	47 (22.1%)	74 (46.5%)
Chronic kidney failure, *n* (%)	86 (23.1%)	61 (28.6%)	25 (15.8%)
Rheumatic disease, *n* (%)	81 (21.8%)	66 (31.0%)	15 (9.4%)
Malignant disease, *n* (%)	69 (18.6%)	49 (23.0%)	20 (12.6%)
Metastatic disease, *n* (%)	26 (7.0%)	20 (9.4%)	6 (3.8%)
Coronary heart disease, *n* (%)	60 (16.1%)	39 (18.3%)	21 (13.2%)
Diabetes mellitus, *n* (%)	59 (15.9%)	32 (15.0%)	27 (17%)
Chronic obstructive pulmonary disease, *n* (%)	49 (13.2%)	16 (7.5%)	33 (20.8%)
Chronic liver insufficiency, *n* (%)	20 (5.4%)	13 (6.1%)	7 (4.4%)
Cerebrovascular disease, *n* (%)	16 (4.3%)	11 (5.2%)	5 (3.1%)
Regular drug abuse, *n* (%)	58 (15.6%)	35 (16.4%)	23 (14.5%)
Past drug abuse, *n* (%)	23 (6.2%)	17 (8.0%)	6 (3.8%)
Suicide attempt in the past, *n* (%)	43 (11.6%)	19 (8.9%)	24 (15.1%)
Domestic violence in the history, *n* (%)	21 (5.6%)	13 (6.1%)	8 (5.0%)

ED = Emergency Department; Results were presented as mean (standard deviation) or median (25th–75th percentile).

**Table 2 jcm-08-00333-t002:** Social state of recurrent ED visitors.

	All Recurrent ED Visitors *N* = 372	Frequent ED Visitors *N* = 213	Repeated ED Visitors *N* = 159
General physician available, *n* (%)	289 (77.7%)	163 (76.5%)	126 (79.2%)
Social homeless, *n* (%)	17 (4.6%)	10 (4.7%)	7 (4.4%)
Decaying condition, *n* (%)	26 (7.0%)	15 (7.0%)	11 (6.9%)
Housing, *n* (%)			
Living alone	116 (31.2%)	60 (28.2%)	56 (35.2%)
Living with others	224 (60.2%)	135 (63.4%)	89 (56.0%)
Supervised living	28 (7.5%)	18 (8.4%)	10 (6.3%)
Unknown	4 (1.1%)	0%	4 (2.5%)
Marital status, *n* (%)			
Single	132 (35.5%)	66 (31.0%)	66 (41.5%)
Married/in partnership	170 (45.7%)	109 (51.2%)	61 (38.4%)
Divorced	57 (15.3%)	28 (13.1%)	29 (18.2%)
Widowed	13 (3.5%)	10 (4.7%)	3 (1.9%)
Children, *n* (%)	189 (50.8%)	114 (53.5%)	75 (47.2%)
Minors	163 (43.8%)	136 (63.8%)	27 (17.0%)
Professional status, *n* (%)			
Employed	117 (31.5%)	59 (27.7%)	58 (36.5%)
Unemployed	56 (15.0%)	34 (16.0%)	22 (13.8%)
In training	19 (5.1%)	12 (5.6%)	7 (4.4%)
Retired	94 (25.3%)	53 (24.9%)	41 (25.8%)
Disabled	86 (23.1%)	55 (25.8%)	31 (19.5%)

ED = Emergency Department.

**Table 3 jcm-08-00333-t003:** Presentation of the recurrent ED patients in the Emergency Department.

	All Recurrent ED Visits *N* = 1921	Frequent ED Visits *N* = 1016	Repeated ED Visits *N* = 905
Number ED visits	4 (4–5)	4 (4–5)	5 (4–6)
Triage by the emergency severity index (ESI), *n* (%)			
ESI 1	16 (0.8%)	11 (1.1%)	5 (0.6%)
ESI 2	148 (7.7%)	75 (7.4%)	73 (8.0%)
ESI 3	1346 (70.1%)	680 (66.9%)	666 (73.6%)
ESI 4/5	411 (21.4%)	250 (24.6%)	161 (17.8%)
Number of ED presentations from,			
*n* (%):			
8 am–5 pm	955 (49.7%)	525 (51.7%)	430 (47.5%)
5 pm–23 pm	496 (25.8%)	254 (25.0%)	242 (26.7%)
11 pm–8 am	470 (24.5%)	237 (23.3%)	233 (25.8%)
Number of self-admittances	4 (3–5)	4 (3–4)	4 (3–5)
Number of admittances by paramedics	0 (0–2)	0 (0–1)	0 (0–2)
Number of emergency allocations by external doctors	1 (0–2)	1 (0–2)	1 (0–2)

ED = Emergency Department; ESI = Emergency Severity Index; Results were presented as median (25th–75th percentile).

**Table 4 jcm-08-00333-t004:** Leading symptoms for repeated visits.

	Repeated ED Visitors *N* = 159
Symptoms due to mental disorders, *n* (%)	45 (28.3%)
With substance abuse, *n* (%)	22 (48.9%)
Symptoms due to gastrointestinal disorders, *n* (%)	34 (21.4%)
Symptoms due to cardiac disorders, *n* (%)	20 (12.6%)
Symptoms due to neurological disorders, *n* (%)	17 (10.7%)
Symptoms due to respiratory disorders, *n* (%)	13 (8.2%)
Symptoms following accidents & complications, *n* (%)	12 (7.5%)
Symptoms due to genito-urinary disorders (incl. nephrology), *n* (%)	9 (5.7%)
Others, *n* (%)	9 (5.7%)

ED = Emergency Department.

**Table 5 jcm-08-00333-t005:** Number of leading symptoms for frequent ED visits.

	Number of Frequent ED Visits *N* = 1016
Symptoms due to gastrointestinal disorders, *n* (%)	176 (17.3%)
Symptoms following head and extremity traumas, *n* (%)	136 (13.4%)
Symptoms due to mental disorders, *n* (%)	105 (10.3%)
Of them with substance abuse, *n* (%)	40 (38.1%)
Symptoms due to neurological disorders, *n* (%)	97 (9.5%)
Symptoms due to cardiac disorders, *n* (%)	95 (9.4%)
Symptoms due to general infection, *n* (%)	94 (9.3%)
Symptoms due to respiratory disorders, *n* (%)	78 (7.7%)
Symptoms due to musculoskeletal disorders, *n* (%)	72 (7.1%)
Symptoms due to complications of previous surgeries, *n* (%)	70 (6.9%)
Symptoms due to genito-urinary disorders (incl. nephrology), *n* (%)	68 (6.7%)
Others, *n* (%)	25 (2.4%)

ED = Emergency Department; General infections were defined as arterial or venous catheter infections, fever of unknown origin or other bacterial or viral infections of unknown origin, cutaneous or subcutaneous abscesses, erysipelas etc.

**Table 6 jcm-08-00333-t006:** Repeated ED Visitors due to mental disorders.

	Repeated ED Visitors Not Because of Mental Disorders *N* = 114	Repeated ED Visitors Due to Mental Disorders *N* = 45	Unadjusted Difference (95% CI, *p*-Value)	Adjusted Difference (95% CI, *p*-Value)
Number of EB visits	5 (4–5)	6 (4–7)	1.5 (0.5–2.5, *p* = 0.004)	1.6 (0.6–2.7, *p* = 0.003)
Number of hospital admissions	2 (0–3)	2 (0–4)	0.5 (−0.3–1.2, *p* = 0.20)	0.9 (0.2–1.6, *p* = 0.011)
Number of day presentations (8 am to 5 pm)	2.5 (1–4)	3 (2–4)	0.05 (−0.7–0.6, *p* = 0.89)	0.1 (−0.6–0.8, *p* = 0.83)
Number of day presentations (5 pm to 11 pm)	1 (0–2)	2 (1–3)	0.7 (0.2–1.2, *p* = 0.010)	0.7 (0.2–1.3, *p* = 0.008)
Number of day presentations (11 pm to 8 pm)	1 (0–2)	2 (1–3)	0.9 (0.3–1.5, *p* = 0.004)	0.9 (0.2–1.5, *p* = 0.008)
			**Unadjusted OR (95% CI, *p*-Value)**	**Adjusted OR (95% CI, *p*-Value)**
Drug abuse in the history, *n* (%)	9 (7.9%)	14 (31.1%)	5.3 (2.1–13.3, *p* < 0.001)	3.9 (1.4–11.1, *p* = 0.010)
Suicide in the history, *n* (%)	7 (6.1%)	14 (31.1%)	9.3 (3.5–24.6, *p* < 0.001)	9.7 (3.3–28.4, *p* < 0.001)
Substance abuse at the ED, *n* (%)	4 (3.5%)	18 (40%)	18.3 (5.7–58.6, *p* < 0.001)	19.6 (5.5–69.5, *p* < 0.001)
Single, *n* (%)	37 (32.5%)	29 (64.4%)	3.8 (1.8–7.8, *p* < 0.001)	3.1 (1.3–7.2, *p* = 0.010)
Out of work, *n* (%)	7 (6.1%)	15 (33.3%)	7.6 (2.9–20.5, *p* < 0.001)	6.7 (2.3–19.5, *p* < 0.001)

ED = Emergency Department; OR = Odds Ratio; CI = Confidence Internal; Results are adjusted for age, sex, Charlson co-morbidity index, presence of general physician; No adjustment was performed if less than five cases occur in both groups.

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
