# Peer review of "Recurrent Emergency Department Users: Two Categories with Different Risk Profiles"

_jcm, 2019, doi:10.3390/jcm8030333_

Round 1
Reviewer 1 Report
Thank you for letting me review this work.
This retrospective study investigates recurrent emergency department presentations (i.e. presenting at least four times) during one year at a tertiary care hospital in Switzerland. The authors group the recurrent presentations into two categories: repeated (presenting with the same symptom(s)) and frequent (presenting due to different symptoms). Differences in characteristics and risk factors for each group are investigated and discussed.
The study, although descriptive in nature, presents interesting and novel data that could add to the literature regarding recurrent emergency department presentations which can significantly influence health care costs and emergency department overcrowding.
Some specific comments and suggestion:
The manuscript would benefit from language editing
Page 2, last paragraph of introduction section: A short explanation of terms such as “one-stop one-shop mentality” could be added
Page 2, methods section, group definition: Were there any recurrent presentations in which the same patient presented with the same symptoms in some cases and with different symptoms in some other cases, and if yes, were these cases grouped as repeated or as frequent?
Page 3, assessment of other variables: Not all of the variables mentioned later in the results and the Tables are presented here, this should be harmonized. Furthermore, a short explanation of the ESI and CCI could be added.
Page 3, results section: The max. number of recurrent presentations should be reported in this section (currently mentioned only later in the discussion). Furthermore, the grouping into repeated and frequent visits is already mentioned in the methods section and should not be repeated in the results. Similarly, sentences such as “The literature has not discriminated within the recurrent ED population…which is a novelty in our study” does not present findings of this study and should therefore not be part of the results section.
Results section: It would be better to harmonize the way results are presented (e.g. report all as n (%))
Results section, Tables 1,2,3: Some of the findings presented in these three Tables could be combined in one Table. Moreover, it would be interesting to present comparisons between the frequent and the repeated ED visitors (for example by adding a column with the p-values)
Results section, Table 4: Table 4a and 4b could be presented together with the repeated and frequent visitis shown next to each other, similar to the previous Tables. This would also facilitate comparisons between the groups, especially by adding p-values
Results section, subgroup of recurrent senior ED patients: The reason for the special interest in this subgroup could be added
Discussion: The novelty of dividinf recurrent presentations into two groups (repeated and frequent) is currently mentioned in several parts of the discussion (and also in the manuscript in general), repetitions should be avoided. Furthermore, the authos report that the finidngs are important for future approaches but do not mention what those could be. This could be added in the discussion.
Author Response
Re-submission of manuscript jcm-451739
Please find attached the revised version of our manuscript entitled “Recurrent Emergency Department Users: Two Categories with Different Risk Profiles”.
We greatly appreciate the chance for revising the manuscript. We have addressed each reviewers’ comment in detail, and revisions are marked by track changes in the manuscript. Please convey our thanks to the reviewer for her/his constructive criticisms and comments.
Reviewer #1
Comment
The manuscript would benefit from language editing
Our reply:
The manuscript was improved and edited by a native English-speaking colleague.
Comment:
Page 2, last paragraph of introduction section: A short explanation of terms such as “one-stop one-shop mentality” could be added
Our reply:
A short explanation of “one-stop one-shop mentality” has been added in the manuscript (lines 89-92).
Comment
Page 2, methods section, group definition: Were there any recurrent presentations in which the same patient presented with the same symptoms in some cases and with different symptoms in some other cases, and if yes, were these cases grouped as repeated or as frequent?
Our reply:
Indeed, in three cases we had to decide the allocation. All three patients had more than six ED visits due to identical health disorders (repeated) but each of them showed one ED visit of different symptoms to the previous visits. Since the majority of the ED visits were in all three cases due to repeated identical symptoms, we therefore decided to group them as repeated ED users.
We added this information in the results section (lines 166 to 169).
Comment
Page 3, assessment of other variables: Not all of the variables mentioned later in the results and the Tables are presented here, this should be harmonized. Furthermore, a short explanation of the ESI and CCI could be added.
Our reply:
We explained ESI and CCI on lines 97-126 and 128-129. Additionally, we improved this paragraph according to reviewer’s comment.
Comment
Page 3, results section: The max. number of recurrent presentations should be reported in this section (currently mentioned only later in the discussion). Furthermore, the grouping into repeated and frequent visits is already mentioned in the methods section and should not be repeated in the results. Similarly, sentences such as “The literature has not discriminated within the recurrent ED population…which is a novelty in our study” does not present findings of this study and should therefore not be part of the results section.
Our reply:
The total number or recurrent ED visits (n = 1,921) was already described at the beginning of the results (line 159). To simplify the reading, we improved the sentence (lines 158 to 160.
The grouping into repeated and frequent ED visitors is a major finding of this study and therefore, we deleted the grouping definition in the methods and presented these findings in the results from line 161 to 166.
We deleted the last sentence “The literature has not…..is a novelty in our study”.
Comment
Results section: It would be better to harmonize the way results are presented (e.g. report all as n (%))
Our reply:
We adjusted all tables according to reviewer’s comment.
Comment:
Results section, Tables 1,2,3: Some of the findings presented in these three Tables could be combined in one Table. Moreover, it would be interesting to present comparisons between the frequent and the repeated ED visitors (for example by adding a column with the p-values)
Our reply:
We would prefer not to combine tables 1-3 in one table because every table is describing different groups of variables: e.g. table 1 presents patients’ characteristics, table 2 is about social factors and table 3 is showing parameters how the ED visitors presented in the ED. To our opinion, it will confuse the reader when pooling these tables into one.
Tests on baseline differences are still common. However, such significance tests assess the probability that observed baseline differences could have occurred by chance.
Therefore, the so-called STROBE (Strengthening the Reporting of Observational Studies in Epidemiology) guidelines state: “significance tests should be avoided in descriptive tables. Also, P values are not an appropriate criterion for selecting which confounders to adjust for in analysis”. According to these epidemiological guidelines, we did not perform such significance tests in Table 1-3.
Comment:
Results section, Table 4: Table 4a and 4b could be presented together with the repeated and frequent visitis shown next to each other, similar to the previous Tables. This would also facilitate comparisons between the groups, especially by adding p-values.
Our reply:
We separated deliberately the symptoms from repeated and frequent ED visitors in two tables to ease the reading for the reader.
Table 4a presents the symptoms in ED users whereas Table 4b shows the reasons of ED visits. Due to different symptoms in every ED visit in the frequent group, we needed to present the symptoms per ED visit and not per ED user.
By pooling both tables into one, the reader might be confused because in one column, the proportion of symptoms in users is presented, and in the next column, the proportion of symptoms in ED visits. Therefore, we prefer to present two tables, 4a and 4b.
Comment:
Results section, subgroup of recurrent senior ED patients: The reason for the special interest in this subgroup could be added
Our reply:
An increased Charlson co-morbidity index (CCI) was shown as a risk factor for multiple ED visits in frequent ED patients. Since senior patients had more co-morbidities and an interaction between increased age and CCI was found, we performed this subgroup analysis in the senior ED population.
We added this explanation (lines 303-304).
Comment:
Discussion: The novelty of dividing recurrent presentations into two groups (repeated and frequent) is currently mentioned in several parts of the discussion (and also in the manuscript in general), repetitions should be avoided. Furthermore, the authors report that the finidngs are important for future approaches but do not mention what those could be. This could be added in the discussion
Our reply:
Thank you for this comment, we deleted some repetitions in the manuscript and added an explanation for future approaches in the discussion (lines 377 to 380).
We have made every effort to address the reviewers’ comments and considerations and believe to have strengthened our manuscript. We also checked our manuscript by a native English-speaking colleague.
We hope the manuscript now fulfills all criteria for a second review.
Yours sincerely
Ksenija Slankamenac, MD PhD
Dagmar I. Keller, MD

Reviewer 2 Report
This is an interesting study by Slenkamenac et al. regarding the analysis of ED recurrent patients. The authors make an interesting and novel distinction of these patients to repeated ED users that always present with the same symptom and frequent ED users that present multiple times per year but with different presenting complaints. The authors try to identify key characteristics of these patient groups that could help improve our understanding regarding the specific etiologies for such behaviors. Such studies could help the design of specific interventions that would minimize this phenomenon and decrease ED overcrowding. The authors also properly identify their study limitations including (most importantly) the fact that they could not have all the ED visits of their reference population.
Although the idea is novel and interesting there are several issues regarding the organization of the manuscript and the data presented that should be further evaluated by the authors. The two more important are the lack of reference population values and the "mixed" language the authors use to describe associations in the form of causations (should be corrected throughout the text). Other issues are:
Major issues
1. This manuscript would benefit greatly by being edited by a professional editing service designed to present the work in scientific English. There were far too many errors to list them all, so all comments below refer to the body of the work.
2. In the Experimental section, the authors should reassess their description of the study population. In essence, their study population was all patients visiting their ED (except<18 years old). From within this population, they identified the patients that reached the study endpoint which were those with four or more ED visits in one year. Actually, the use of endpoints in such a descriptive study is rather misleading. I therefore suggest an extensive re-writing of the study design description in order to clarify such issues.
3. There is a clear need for the authors to provide more information regarding the reference population. For example, what are the Table 1 and Table 2 characteristics (in % ) of the non-recurrent patients. Such information is critical for the identification of patients in the general population that have the potential to become recurrent ED visitors (or members of one of the suggested subgroups). To what extend do patients that do not become recurrent ED visitors differ from those who become? Proper likelihood ratios for such behaviors should be provided. In its current form, the study provides some new insight for ED physicians but this is limited due to the lack of the reference population data. "ED Frequent Flyers" will always be older, more comorbid, more mentally compromised, poorer and with less supportive environment. Identifying those who will become recurrent ED visitors would be of significant importance for ED physicians and could help design interventions to reduce such behaviors.
4. The group of repeated ED visitors is reported to be more likely to present with COPD symptoms (lines 221-2) yet the authors report only 13 patient in this group in the text and 33 patients in table 1. Please clarify this discrepancy. Is it possible that in table 1 instead of visitors the authors describe visits?
Minor issues
1. Lines 251-252. A logical loophole is described since "frequent ED visitors being homeless or unemployed showed a trend for increased risk for recurrent ED visits"
Author Response
Re-submission of manuscript jcm-451739
Please find attached the revised version of our manuscript entitled “Recurrent Emergency Department Users: Two Categories with Different Risk Profiles”.
We greatly appreciate the chance for revising the manuscript. We have addressed each reviewers’ comment in detail, and revisions are marked by track changes in the manuscript. Please convey our thanks to the reviewer for her/his constructive criticisms and comments.
Reviewer #2
Comment
This manuscript would benefit greatly by being edited by a professional editing service designed to present the work in scientific English. There were far too many errors to list them all, so all comments below refer to the body of the work.
Our reply:
The manuscript was improved and edited by a native English-speaking colleague.
Comment:
In the Experimental section, the authors should reassess their description of the study population. In essence, their study population was all patients visiting their ED (except<18 years old). From within this population, they identified the patients that reached the study endpoint which were those with four or more ED visits in one year. Actually, the use of endpoints in such a descriptive study is rather misleading. I therefore suggest an extensive re-writing of the study design description in order to clarify such issues.
Our reply:
We improved the method section and especially concentrated on the improvement and clarification of the study design. Please find the changes marked by “track changes” in the revised manuscript (lines 70 -92).
Comment
There is a clear need for the authors to provide more information regarding the reference population. For example, what are the Table 1 and Table 2 characteristics (in %) of the non-recurrent patients. Such information is critical for the identification of patients in the general population that have the potential to become recurrent ED visitors (or members of one of the suggested subgroups). To what extend do patients that do not become recurrent ED visitors differ from those who become? Proper likelihood ratios for such behaviors should be provided. In its current form, the study provides some new insight for ED physicians but this is limited due to the lack of the reference population data. "ED Frequent Flyers" will always be older, more comorbid, more mentally compromised, poorer and with less supportive environment. Identifying those who will become recurrent ED visitors would be of significant importance for ED physicians and could help design interventions to reduce such behaviors.
Our reply:
Thank you for the suggestion to add the non-recurrent ED population as a reference.
In many studies, the general ED populations has been described and compared to the recurrent ED users. There will not be any new information or benefit adding the information about the general population.
But none of the published studies have investigated within the recurrent ED patients. Therefore, the focus of this study was on the subgroup of recurrent ED patients and not in the proportion to the general non-recurrent ED population.
27’998 ED patients were evaluated in this current study. Due to our focus on the recurrent ED population (N = 372), it would be disproportionate to complete the parameter of the remaining 27’626 ED patients in an appropriate period without any prospect of novelty.
Nevertheless, we may currently provide some information about the general and non-recurrent ED population. The non-recurrent ED population was in mean 45 yrs. (SD 19) old, 54% of them were male. After tests and first treatment in the ED, every fourth patient (25.7%) needed to be admitted to the hospital. 43.4% of patients visited the ED due to consequences of an injury or trauma whereas the remaining 56.6% had medical problems including poisoning, drugs and alcohol intoxications and psychological disorders. Most of our patients (63.4%) were triaged into the Emergency Severity Index (ESI) category 3, followed by 29.6% as ESI 4, 6.5% as ESI 2 and 0.5% as ESI 5.
We included these general data in the results section on p. 3, lines 170-175.
Comment
The group of repeated ED visitors is reported to be more likely to present with COPD symptoms (lines 221-2) yet the authors report only 13 patients in this group in the text and 33 patients in table 1. Please clarify this discrepancy. Is it possible that in table 1 instead of visitors the authors describe visits?
Our reply:
Thank you for the attentive reading. The 13 patients suffered from respiratory disorders as repeated leading symptoms. We corrected that in the lines 260-263 instead of chronic obstructive pulmonary disease.
In Table 1, we described solely the co-morbidities of the recurrent ED populations and in 33 patients, a COPD was reported and did not cause absolutely the ED presentation.
Comment (minor issue)
Lines 251-252. A logical loophole is described since "frequent ED visitors being homeless or unemployed showed a trend for increased risk for recurrent ED visits"
Our reply:
Indeed, homeless and unemployment are known risk factors for recurrent ED visitors in the literature but in Switzerland, it is unusual that homeless people use the ED for sleeping accommodations without any medical problem.
Thus, we could show a trend that those two factors may be risk factors in the frequent ED patients. Whereas in the group of repeated ED patients, homeless and unemployment was not identified as risk factors, not even a trend was shown.
We improved the sentence (lines 295 to 296).
We have made every effort to address the reviewers’ comments and considerations and believe to have strengthened our manuscript. We also checked our manuscript by a native English-speaking colleague.
We hope the manuscript now fulfills all criteria for a second review.
Yours sincerely
Ksenija Slankamenac, MD PhD
Dagmar I. Keller, MD

Round 2
Reviewer 1 Report
Thank you for providing a revised manuscript and for addressing all prior
comments and concerns.
There are also some minor corrections/editing need to be addressed (e.g. Table 2: "Results are
presented as mean (standrad deviation) or median (25th-75th percentile)"
not applicable any more, and some other similar minor text edits).
Author Response
Re-re-submission of manuscript jcm-451739
Please find attached the revised version of our manuscript entitled “Recurrent Emergency Department Users: Two Categories with Different Risk Profiles”.
We greatly appreciate the chance for revising the manuscript a second time. We have addressed each reviewers’ comment in detail, and revisions are marked by track changes in the manuscript. Please convey our thanks to the reviewer for her/his constructive criticisms and comments.
Reviewer #1
Thank you for providing a revised manuscript and for addressing all prior comments and concerns.
Comment 1:
There are also some minor corrections/editing need to be addressed (e.g. Table 2: "Results are presented as mean (standard deviation) or median (25th-75th percentile)" not applicable any more, and some other similar minor text edits).
Our reply:
Thank you for your comment. We performed additional edits according to reviewer’s comment.
Please find the changes marked by “track changes” in the re-revised manuscript.
We have made every effort to address the reviewers’ comments and considerations and believe to have strengthened our manuscript. We checked additionally the English language and performed minor changes.
Sincerely yours,
Ksenija Slankamenac, MD PhD
Dagmar I. Keller, MD

Reviewer 2 Report
This is a revised version of the manuscript by Slankamenac et al. that has improved significantly and has responded appropriately to most of my previous points.
There are still some mistakes in English (e.g. lines 70-71: " We enrolled all patients of the ED of a tertiary care hospital from January 1st, 2015 to December 31st, 2015 and who visited the ED at least four times during a one-year period."). Such minor mistakes exist in several points where the track changes tool has been used and can be easily corrected with a careful reading of a clean version.
My second comment regarded the description of the methodology and has been addressed accordingly.
My third comment regarding the inclusion in the manuscript of data regarding the non-recurrent ED visitors has also been addressed to some extent. The authors provide some info, although not as much as it would be significantly useful for the educated reader. I believe that the explanation provided by the authors regarding the disproportionate effort needed to complete these parameters is reasonable.
My fourth comment was about a possible discrepancy between the number of COPD patients reported in table 1 and in the text of the original version. The authors have explained the reason for this discrepancy (although in their response they did not refer to the proper corrected text (they pointed to 260-263 instead of 230-234)). A small comment regarding the fact that in table 1 they solely describe the co-morbidities of the recurrent ED population, needs to be included in the text or the table legend.
My fifth comment has been addressed properly.
Author Response
Re-re-submission of manuscript jcm-451739
Please find attached the revised version of our manuscript entitled “Recurrent Emergency Department Users: Two Categories with Different Risk Profiles”.
We greatly appreciate the chance for revising the manuscript a second time. We have addressed each reviewers’ comment in detail, and revisions are marked by track changes in the manuscript. Please convey our thanks to the reviewer for her/his constructive criticisms and comments.
Reviewer #2
This is a revised version of the manuscript by Slankamenac et al. that has improved significantly and has responded appropriately to most of my previous points.
Comment 1:
There are still some mistakes in English (e.g. lines 70-71: " We enrolled all patients of the ED of a tertiary care hospital from January 1st, 2015 to December 31st, 2015 and who visited the ED at least four times during a one-year period."). Such minor mistakes exist in several points where the track changes tool has been used and can be easily corrected with a careful reading of a clean version.
Our reply:
Thank you for your comment. We performed additional adjustments and edited it once again.
Please find the changes marked by “track changes” in the re-revised manuscript.
Comment 2:
My second comment regarded the description of the methodology and has been addressed accordingly.
Our reply:
Thank you.
Comment 3:
My third comment regarding the inclusion in the manuscript of data regarding the non-recurrent ED visitors has also been addressed to some extent. The authors provide some info, although not as much as it would be significantly useful for the educated reader. I believe that the explanation provided by the authors regarding the disproportionate effort needed to complete these parameters is reasonable.
Our reply:
Thank you.
Comment 4:
My fourth comment was about a possible discrepancy between the number of COPD patients reported in table 1 and in the text of the original version. The authors have explained the reason for this discrepancy (although in their response they did not refer to the proper corrected text (they pointed to 260-263 instead of 230-234)). A small comment regarding the fact that in table 1 they solely describe the co-morbidities of the recurrent ED population, needs to be included in the text or the table legend.
Our reply:
We commented to the lines 221-222 of the first submitted manuscript. Unfortunately, due to the changes during the revision process, these lines appeared in the revised manuscript in position 260-263.
We added the co-morbidities in the title of table 1 (line 198) and in the text on line 183 according to reviewer’s comment.
Comment 5:
My fifth comment has been addressed properly.
Our reply:
Thank you.
We have made every effort to address the reviewers’ comments and considerations and believe to have strengthened our manuscript. We checked additionally the English language and performed minor changes.
Sincerely yours,
Ksenija Slankamenac, MD PhD
Dagmar I. Keller, MD
